# Thin Niobium and Niobium Nitride PVD Coatings on AISI 304 Stainless Steel as Bipolar Plates for PEMFCs

**Masoud Atapour** [1,2,*], **Vahid Rajaei** [1], **Stefano Trasatti** [2], **Maria Pia Casaletto** [3] and **Gian Luca Chiarello** [4]

1 Department of Materials Engineering, Isfahan University of Technology, Isfahan 84156-83111, Iran; rajaei.vahid@gmail.com
2 Department of Environmental Science and Policy, Università degli Studi di Milano, via Celoria 2, 20133 Milano, Italy; stefano.trasatti@unimi.it
3 Istituto per lo Studio dei Materiali Nanostrutturati, Consiglio Nazionale delle Ricerche, Via Ugo La Malfa 153, 90146 Palermo, Italy; mariapia.casaletto@cnr.it
4 Department of Chemistry, Università degli Studi di Milano, via Golgi 19, 20133 Milano, Italy; gianluca.chiarello@unimi.it
* Correspondence: m.atapour@iut.ac.ir; Tel.: +98-313-3915735

**Abstract:** In this paper, Nb, NbN, and Nb/NbN thin films were successfully deposited on AISI 304 stainless steel (304 SS) as the bipolar plate (BPP) for proton-exchange membrane fuel cell (PEMFC) by employing a radio-frequency (RF) magnetron sputtering system. Corrosion assessments in simulated PEMFC operating conditions (1 M $H_2SO_4$ + 2 mg/kg HF, 70 °C) revealed that the Nb and NbN coatings significantly improved the corrosion resistance of the 304 SS substrates. The Nb and NbN deposited samples at 350 °C exhibited superior corrosion resistance compared to those coated at 25 °C. Potentiostatic tests were also performed at the constant potentials of +0.644 and −0.056 V vs. Ag/AgCl to simulate the cathodic and anodic PEMFC conditions, respectively. The minimum current densities were recorded for the Nb coating in both anodic and cathodic conditions. Compared with the 304 SS substrate, all coatings showed lower interfacial contact resistance (ICR) and higher hydrophobicity. Among the tested coatings, the Nb coating exhibited the smallest ICR (9 mΩ·cm$^2$ at 140 N/cm$^2$). The results of this investigation revealed that the Nb and NbN coatings deposited by RF magnetron sputtering on 304 SS can be regarded as promising candidates for BPPs in PEMFCs.

**Keywords:** RF sputtering PVD coating; NbN coating; Nb coating; corrosion; metallic bipolar plate; proton exchange membrane fuel cell

## 1. Introduction

There have been many attempts to optimize fuel utilization and minimize greenhouse gas emissions. Moreover, global interest in having alternative energy sources for fossil fuels is continuously increasing due to the escalation of oil prices, growing energy demand, and global climate changes [1]. Polymer electrolyte membrane fuel cells (PEMFCs) have attracted considerable attention for their stationary, automotive, and portable applications [2]. PEMFCs offer the advantages of high efficiency, low operating temperature, eco-friendly behavior, and easy start-up [3]. However, the high manufacturing cost and relatively high weight of PEMFCs have limited the potential use of this kind of energy systems in different applications [4,5]. Based on previous reports, bipolar plates (BPPs) account for about 80% of the total mass and 45% of the total stack cost [6]. Therefore, it is important to develop light and low-cost materials for BPPs to make PEMFC more suitable for practical services

Graphite-based bipolar plates BPPs, as traditional options, have excellent corrosion resistance and good electrical conductivity. However, these BPPs are not practical candidates for transportation due to their poor mechanical properties, high manufacturing costs, and poor machinability [7].

Stainless steels (SS) can be an ideal material for BPPs owing to their excellent mechanical properties, high electrical conductivity, and reasonable price [8]. However, SSs are susceptible to undergo corrosion attacks in acidic conditions of a fuel cell and poison the membrane due to the release of metal ions (e.g., Fe, Cr, and Ni). Moreover, the formation of a passive oxide film on the surface of SSs increases the electrical resistivity of the BPPs [6].

To get over these problems, many attempts have been performed by applying different surface modifications for SSs. Many coatings based on the transition metals and their nitrides were developed using different techniques to improve the anti-corrosion behavior of the stainless steel BPPs [9].

Recently, niobium-based coatings have received widespread acceptance in the field of BPPs [10,11] due to their excellent corrosion resistance in mineral acids environments [11,12], like sulphuric acid (the main component of the electrolyte within the PEMFC stacks), and high electrical conductivity coupled with good mechanical properties. Pozio et al. [13] fabricated a niobium clad material on the surface of the 430 SS and reported promising corrosion properties for BPPs applications. However, it has been reported that the optimum thickness of the cladded layers can be a big challenge [14]. In a work by Kim et al. [15], a magnetron sputtered niobium thin film was investigated as an alternative to the niobium cladded material on stainless steel BPPs. They concluded that the sputtered niobium coating exhibited several advantages over the niobium cladded layer. In other works [16], Dadfar et al. prepared a niobium based coating using thermo-reactive deposition (TRD) on a plasma nitrided AISI 304 Stainless Steel (304 SS) and reported that low ICR values and very good corrosion resistance can be obtained after a pickling treatment by 20% $HNO_3$ and 5% HF. Liang et al. [17] reported that the electro-deposition of Nb on 304 SS remarkably improved corrosion resistance in the simulated PEMFC environments due to the formation of niobium oxides on the surface. Similar observations were reported by Cui et al. [18] on the niobium modified 316 L via the plasma surface diffusion alloying (PSDA) method. Lin et al. [19] developed a hybrid plasma surface treatment to simultaneously coat alloy 316 SS surfaces with both N and Nb. They reported that the ICR values of the hybrid plasma-treated 316 SS specimens were significantly lower than the target value of the US Department of Energy. In addition, their results signify that the duplex-layer coating is superior to the single-layer one for BPPs.

Cha et al. [20] focused on the surface treatment of 304 SS by sputtering Nb and Cr with $N_2$ as a reaction gas. They found that the corrosion resistance of NbN/NbCrN multiphase films was higher than the NbN coatings and bare 304 SS. However, the ICR values of NbN/NbCrN and NbN coatings were higher than the target value of the US Department of Energy. Feng et al. [21] investigated the niobium implanted 316 L for BPP applications and reported that the niobium implantation can significantly improve the corrosion resistance and the electric conductivity of the 316 L SS. Similar results were reported for the niobium implanted 430 SS [13]. It is worth mentioning that, although these ion implantations improved the corrosion resistance of the stainless steels, the ICR values of the coated specimens were still much higher than the target value of the US Department of Energy (10 m$\Omega$·cm$^2$) due to the formation of Nb oxides on the surface of the substrate [22].

Compared to the mentioned coating treatments, such as cladding, ion implantation, and plasma-enhanced reactive evaporation, the magnetron sputtering technique is reported to be a simple and inexpensive method for producing coatings on BPPs [9]. An attempt was made by Singh et al. [23] to develop NbN coatings on stainless steel by reactive DC magnetron sputtering. They observed that the hardness, adhesion, and corrosion resistance of the coatings were improved remarkably after incorporation of a 10 μm thick Cr interlayer between the NbN and substrate. Fonseca et al. [24] reported that the Al addition to NbN can improve the corrosion resistance of the magnetron sputtered coating on 316 SS. In a recent work by Shi et al. [25], magnetron sputtered TiNb and TiNbN thin films were investigated, and concluded that these thin films can improve corrosion resistance and

electrical contact conductivity of 316 L BPPs. A literature survey on the development of coatings for BPP applications revealed that multiphase and/or multilayer coatings are desirable and special attention has been given to the development of these coatings [26,27].

RF magnetron sputtering has been demonstrated as an appropriate method in making porosity-free coatings [26]. Materials such as WC, $Si_3N_4$, $Al_2O_3$, and TiN can all be sputtered by employing the RF magnetron sputtering technique to give very adherent hard coatings with thicknesses up to several micrometers [28]. Jun et al. [29] developed NbN films on AISI 304 SS using inductively coupled plasma (ICP) assisted DC magnetron sputtering method and reported that it is possible to obtain excellent wear and corrosion properties for the films by this technique. However, they did not report the ICR and wettability of the NbN coatings. Furthermore, the influence of deposition of a pure Nb monolayer was not addressed.

This work aimed to develop the Nb, NbN, and Nb/NbN coatings for BPP applications on 304 SS by employing a radio frequency (RF) magnetron sputtering system available at the Università Degli Studi Di MilanoDifferent characterizations of SEM, XRD, XPS, ICR, wettability, and electrochemical evaluations of potentiodynamic polarization and potentiostatic polarization tests were carried out on the coated and 304 SS uncoated specimens.

## 2. Materials and Methods

### 2.1. Specimen Preparation

Nb, NbN, and Nb/NbN thin coatings were deposited on $10 \times 10$ cm$^2$ 304 SS plates using a reactive RF magnetron sputtering system (Cinquepascal S.r.l., Trezzano S/N, Italy). It needs to be noted that RF magnetron sputtering was used in this work as a beneficial strategy to improve the coating microstructure. The coatings for thickness measurements and XPS examinations were also made on 5 $\times$ 10 mm$^2$ silicon wafer samples attached to the surface of the SS plates. Before the coating process, the stainless steel samples were ground with SiC papers up to #1200 grit and finally polished using alumina (0.25 μm particle size) slurry. The samples were then cleaned with acetone, ethanol, and deionized water in an ultrasonic cleaner (with the power of 50 W) for 20 min, and finally dried before deposition. For comparative purposes, this surface preparation was conducted according to previous papers [16]. A schematic of the RF magnetron sputtering system and the applied parameters used in this work are shown in Figure 1 and Table 1, respectively. The stainless steel substrates were placed at a distance of 12 cm above the metallic niobium target (50.8 mm diameter, Good fellow Gmbh, Germany). The substrate holder was rotated with a rotation of 5 rpm for 30 min. Before the deposition process, the chamber was evacuated at a pressure of ca. $5 \times 10^{-6}$ mbar. The coating process was performed at two substrate temperatures of 25 °C (rising up to ca. 50 °C during the deposition due to the heating effect of the ion bombardment) and 350 °C. The pressure was then raised to 3 Pa with pure Ar (for metallic Nb deposition) or 20% $N_2$/Ar mixture (for NbN deposition). The target was sputtered for 15 min with the shutter closed, before starting the deposition to remove eventual surface contaminations.

**Table 1.** Radio frequency (RF) sputtering parameters used for the deposition of Nb, NbN, and Nb/NbN coatings.

| Target | Nb |
| --- | --- |
| Substrate | 304 SS |
| RF Power (W) | 150 |
| DC Self Bias (V) | 200 |
| Gas composition | 20% $N_2$/Ar (or pure Ar) |
| Deposition Time | 2 h for Nb and 4 h for NbN |
| Substrate temperature (°C) | 25 and 350 |
| Pressure (Pa) | 3 |

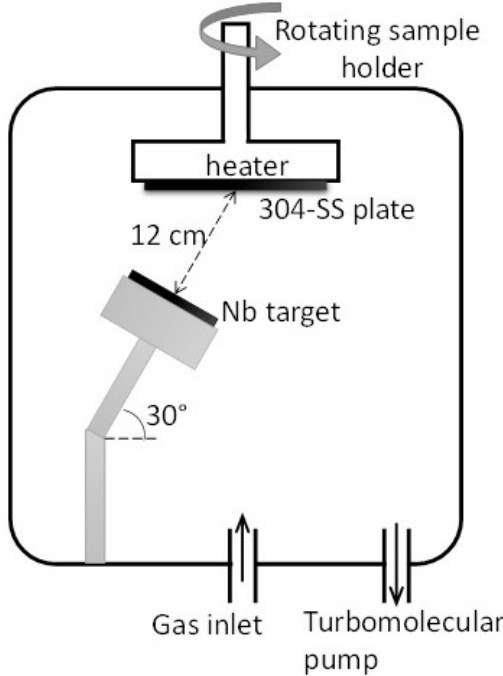

**Figure 1.** Schematic representation of the reactive Radio Frequency (RF) magnetron sputtering used for Nb and NbN coating deposition.

Deposition parameters and conditions were adjusted to obtain coatings with almost the same thickness. Table 2 summarizes the labels and PVD coating conditions of each sample.

**Table 2.** Sample labels and PVD coating conditions of each sample.

| | |
|---|---|
| **Nb-HT** | One layer of Nb coating, deposited at 350 °C and 3 Pa of pure Ar for 2 h |
| **NbN-RT** | One layer of NbN coating, deposited at 25 °C and 3 Pa of 20%$N_2$/Ar for 4 h |
| **NbN-HT** | One layer of NbN coating, treated in 350 °C and 3 Pa of 20%$N_2$/Ar for 4 h |
| **Nb/NbN-HT** | Two layers consisting of Nb (deposited at 350 °C for 2 h and pressure of 3 Pa of pure Ar) and NbN (deposited at 350 °C for 4 h and pressure of 3 Pa of 20%$N_2$/Ar) |

*2.2. Surface Characterizations*

The surface morphology of the coated specimens was investigated by using Scanning Electron Microscopy (SEM, Zeiss LEO 1430 microscope (Zeiss, Jena, Germany) at 1000× magnification. X-ray Diffraction (XRD) studies were carried out with a PW3020 powder diffractometer (Philips, Amsterdam, The Netherlands) using Cu K$\alpha$ radiation. The analysis was conducted at a Bragg angle range of 30° < 2$\theta$ < 80°.

X-ray Photoelectron Spectroscopy (XPS) was used to characterize the surface composition of pure Nb and niobium nitride PVD coatings on 304 SS substrates. XPS analysis was performed using a VG Microtech ESCA 3000 Multilab spectrometer (VG Scientific Ltd., East Grinstead, UK), equipped with a standard Al K excitation source (h$\nu$ = 1486.6 eV) and a multi channeltron detection system. The base pressure in the spectrometer chamber was lower than $1 \times 10^{-6}$ Pa. The binding energy (BE) scale was calibrated using the C 1 s signal from the surface contamination (BE = 285.1 eV). The accuracy of the energy measure was ±0.1 eV.

XPS data were analyzed by XPS peak software using a nonlinear least square curve-fitting procedure. Surface relative atomic concentrations were calculated by a standard quantification routine, including Wagner's energy dependence of attenuation length and a standard set of VG Escalab sensitivity factors. The uncertainty of the atomic quantitative analysis was about ±10%.

The surface roughness ($R_a$) and the thickness of the deposited thin films were measured with a stylus profilometer (Bruker Dektak XT, Tucson, AZ, USA). These experiments were repeated at least five times and the mean values were reported. It needs to be mentioned that the thickness of the coatings was determined using the coated silicon wafers. In addition, the surface wettability of different specimens was evaluated by measuring the contact angle (θ) on the surface at ambient temperature and atmospheric pressure. The wettability of different specimens was recorded using a Krüss Easy Drop instrument (Hamburg, Germany) and analyzed using TS view software. The measurements were conducted using a 2 μL water droplet in static condition. The wettability investigation was conducted at five different locations on each sample and the average values were reported.

### 2.3. Electrochemical Measurements

The electrochemical assessments were carried out using DC potentiodynamic and potentiostatic polarization techniques. The corrosion tests were accomplished using a conventional three-electrode cell containing an Ag/AgCl reference electrode, a platinum counter electrode (Pt mesh), and a working electrode (the coated and 304 SS uncoated specimens with an exposed area of 1 cm$^2$) by a Gamry reference 600 instrument (Gamry Instruments, Warminster, UK). To avoid chloride contaminations, the Ag/AgCl reference electrode was placed in a separate compartment containing a solution of saturated $K_2SO_4$ and was connected to the cell using a lugging capillary filled with one molar sulfuric acid solution.

To simulate fuel cell environments, all electrochemical examinations were carried out in a solution containing 1 M $H_2SO_4$ + 2 mg/kg HF at 70 °C [16].

All electrochemical tests were started after 30 min of immersion at the open circuit potential (OCP). The polarization tests were carried out from −250 mV vs. OCP to +1.0 V vs. Ag/AgCl at a constant scan rate of 0.32 mV s$^{-1}$. Potentiostatic measurements were conducted for 30 min at constant potentials of −0.05 and +0.644 V to simulate the anodic and cathodic PEMFC conditions, respectively. Each experiment was repeated at least three times to ensure repeatability of results.

### 2.4. Interfacial Contact Resistance (ICR) Measurements

To obtain the ICR of the bare and coated specimens, the method described by Wang et al. [16] was pursued. Specimens were sandwiched between two pieces of carbon papers and then they were put between two gold-coated copper plates for pressure loading. Subsequently, a constant electrical current of 1 A was passed through the copper plates and the voltage drop was measured by a NIMEX-NI3310 multimeter (Shanghai Handsun Electronic Co., Ltd., Shanghai, China) at a compaction force of 140 N/cm$^2$. The contact resistance value was calculated based on Ohm's law. The needed compaction force was applied by weight and potentials were recorded.

### 2.5. Statistical Analysis

Data are reported as mean values plus standard deviation (SD). A student's t-test for unpaired data with unequal variance (KaleidaGraph v.4.0) was applied to evaluate whether differences between the two datasets were statistically different. *p* values < 0.05 were considered to show significance. The word "significant" is only used for statistically significant differences in this paper.

## 3. Results

### 3.1. Surface Morphology and Composition

The surface morphology of different coatings is shown in Figure 2. As it can be seen, all of the prepared coatings are uniform and relatively compact. Moreover, surface morphologies were relatively similar in all deposited coatings. All the coatings exhibited a regular grain-like morphology, which was the typical morphology of thin films prepared by RF-magnetron sputtering [30]. Though the top surface images reveal that the morphology of the NbN coating deposited at 350 °C (Figure 2c,d) is

similar to that obtained at room temperature, it should be noted that Cakir et al. [31] suggested that the temperature increase of the substrate surface during PVD process can affect the reaction kinetics. As a consequence of that, layers deposited at high substrate temperatures could have higher density due to the smaller energy loss of sputtered atoms [32]. Therefore, it could be expected a similar effect must also be established in our samples, but this is not easily distinguished by a conventional SEM examination. However, it is proven that other factors such as target composition, $N_2$ and Ar gas flows, surface preparation, evolved texture of films and the formation of residual stress in the coatings are important factors influencing the variation of density and microstructure [33].

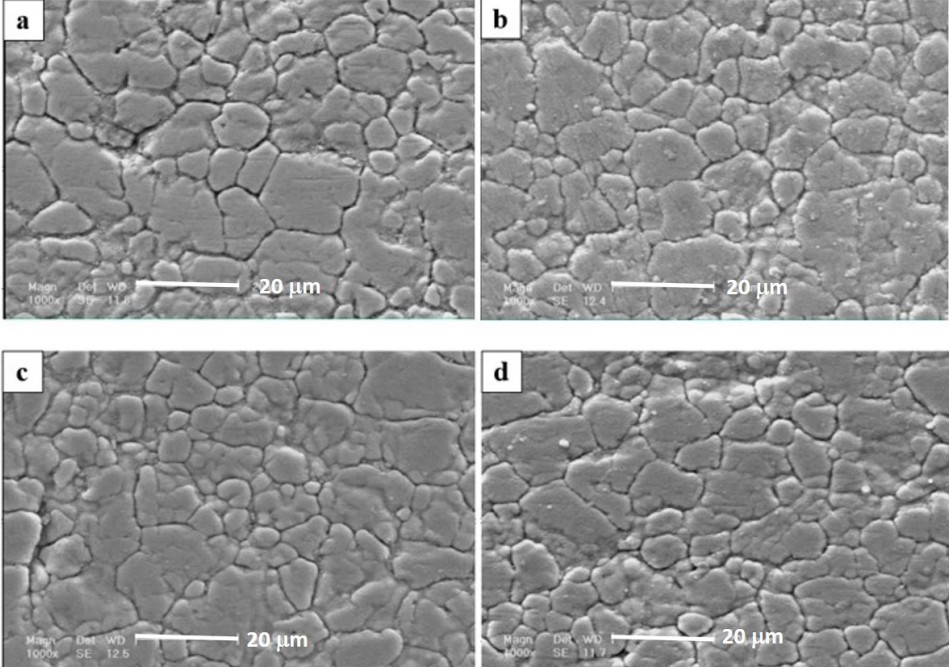

**Figure 2.** SEM images showing the surface morphology of (**a**) Nb-HT, (**b**) NbN-RT, (**c**) NbN-HT, (**d**) Nb/NbN-HT.

The thickness and roughness of the investigated coatings measured by contact profilometer are collected in Table 3. It can be seen that the NbN-HT sample shows a higher surface roughness (0.05 μm) compared to the corresponding sample deposited at RT (0.03 μm). These observations are in good agreement with that of CrN films deposited by RF magnetron sputtering [32]. Among all the tested coatings, the pure Nb layer exhibited the lowest thickness (0.75 μm) and roughness (ca. 0.01 μm).

**Table 3.** Thickness and roughness of PVD-Nb-HT, PVD-NbN-RT, NBN-HT, and PVD-Nb/NbN-HT PVD coatings deposited on 304 SS.

| Coating | Coating Thickness (μm) | Roughness (μm) |
|---|---|---|
| Nb-HT | 0.75 ± 0.05 | 0.011 ± 0.001 |
| NbN-RT | 1.11 ± 0.04 | 0.029 ± 0.005 |
| NbN-HT | 1.13 ± 0.01 | 0.05 ± 0.01 |
| Nb/NbN-HT | 1.8 ± 0.1 | 0.04 ± 0.01 |

The typical X-ray diffraction patterns of the 304 SS substrate and different deposited thin films are present in Figure 3. The diffractogram of Nb coating (Figure 3a) revealed the characteristic peaks of bcc Nb at 2θ = 38° and 69° due to the (110) and (211) crystal planes of pure niobium as well as those at 2θ = 44°, 45°, 51° and 75° of γ-iron (111), α-iron (110), γ-iron (200) and γ-iron (220), respectively.

The detection of iron from the substrate leads to the observation that the thickness of the niobium thin film was lower than the X-ray penetration depth. The XRD patterns of the NbN thin films prepared at room temperature and 350 °C revealed similar characteristic peaks at 2θ = 36°, 42°, 61°, and 72°, which were related to the (111), (200), (220), and (311) crystal planes of NbN, respectively. Also, it can be deduced that the NbN thin films formed at low and high temperatures exhibited a single phase of face-centered cubic (FCC) structure [34]. However, the XRD pattern recorded for the NbN coating deposited at 350 °C (NbN-HT) exhibited more intense peaks than those recorded for the NbN-RT. This result confirms that higher crystallinity was obtained at higher deposition temperature.

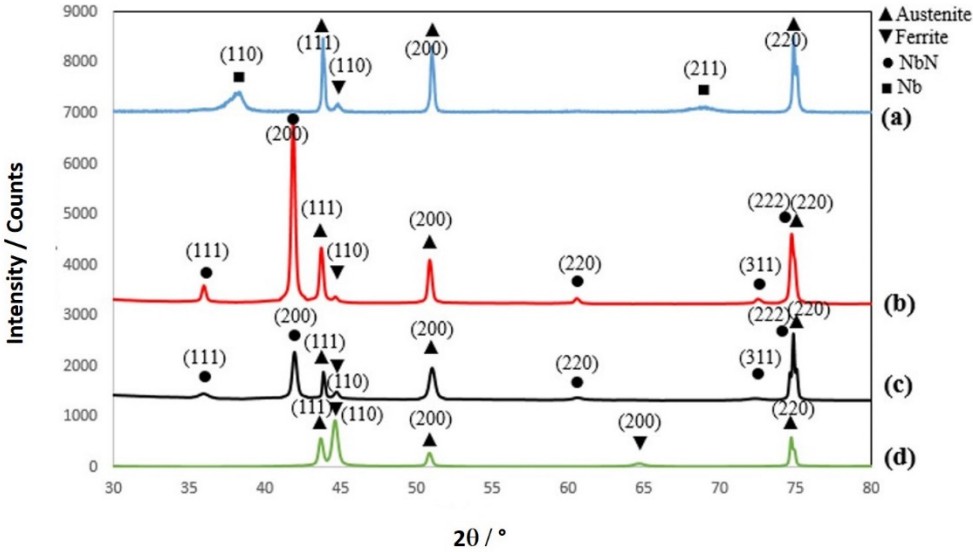

**Figure 3.** XRD patterns of (**a**) Nb-HT, (**b**) NbN-HT, (**c**) NbN-RT, and (**d**) 304 SS substrate.

Figure 4 shows the XRD spectra of PVD-Nb/NbN multilayer coating. It indicates that the Nb/NbN coating exhibited a XRD pattern very similar to that observed for the NbN as deposited at high temperatures. It is interesting to notice that the relative intensity of the (200) crystal plane in all NbN coatings is higher than the reference NbN, indicative of a preferred orientation of the crystallites grown on the 304 SS substrate.

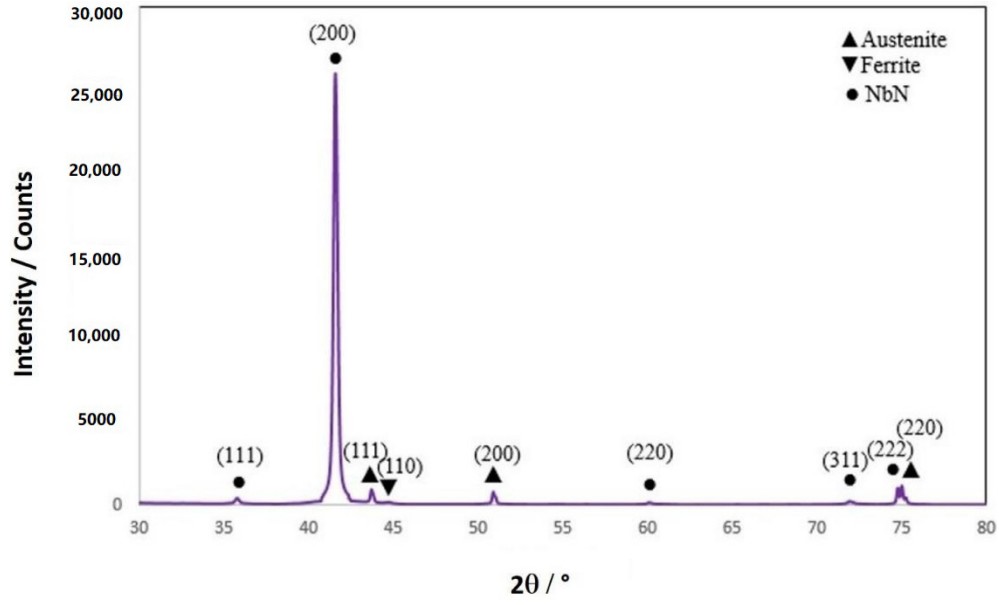

**Figure 4.** XRD pattern of Nb/NbN-HT coating deposited on 304 SS substrate at a high temperature.

### 3.2. XPS Surface Analysis

XPS spectra were recorded for different coatings deposited on Si wafers (Figure 5). The relative surface chemical compositions were summarized in Table 4. As expected, the PVD-NbN monolayer and PVD-Nb/NbN multilayer coatings deposited at high temperature exhibited a very similar surface chemical composition. It can be seen that the highest concentration of niobium and the lowest extent of nitrogen were detected in the PVD-NbN coating prepared at room temperature (PVD-NbN-RT).

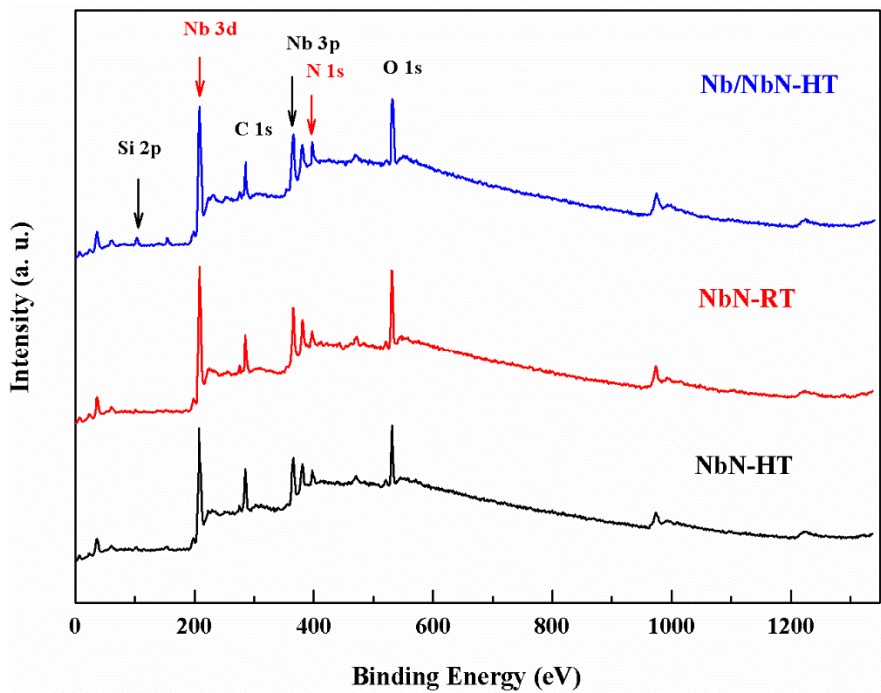

**Figure 5.** XPS survey spectra results of different coatings deposited on Si wafers.

**Table 4.** XPS relative surface chemical composition of the coatings, expressed as an atomic percentage (at.%).

| Sample | Nb 3d | N 1s | O 1s | Si 2p | *N/Nb* | *O/Nb* |
|--------|-------|------|------|-------|--------|--------|
| NbN-RT | 33.2 | 14.6 | 46.3 | 5.9 | 0.44 | 1.39 |
| NbN-HT | 28.1 | 18.7 | 38.1 | 15.1 | 0.66 | 1.36 |
| Nb/NbN-HT | 27.5 | 16.9 | 39.3 | 16.3 | 0.61 | 1.43 |

The XPS spectrum of Nb 3d consisted of Nb $3d_{5/2}$ and Nb $3d_{3/2}$ doublet peaks. The curve-fitting of the Nb 3d spectra of all investigated samples were decomposed into three doublet components, as shown in Figure 6. The lowest binding energy doublet (Nb $3d_{5/2}$ at BE = 204.7–205.1 eV) corresponded to the $NbN_x$ and/or NbO species. The second doublet (Nb $3d_{5/2}$ at 206.2 eV) was associated with the oxynitrideoxinitride $NbN_xO_y$ species. Finally, the higher binding energy doublet (Nb $3d_{5/2}$ at 207.8 eV) can be related to the $Nb_2O_5$ [35]. The surface distribution of niobium species obtained from the curve-fitting of XPS Nb 3d spectra (total peak area = 100%) is presented in Table 5.

Furthermore, the XPS analyses confirmed that the $NbN_x$ surface layer was probably contaminated by oxygen. The PVD-Nb/NbN multilayer coating and the NbN monolayer (PVD-NbN-RT) exhibited the highest and lowest niobium oxygenated species, respectively. It seems the oxygen contamination occurred when the $NbN_x$ film was exposed to ambient air. It has been also reported that a native $Nb_2O_5$ oxide layer grew on NbN when exposed to air [36,37]. However, the presence of oxygen detected by XPS in our experimental conditions corresponded only to a surface composition, since no evidence of $Nb_2O_5$ crystalline phase in the bulk could be inferred by XRD analysis. The distribution of oxygen

species on the surfaces extracted from the curve-fitting of XPS O 1s spectra (total peak area = 100%) and reported in the lowest concentration of Nb oxidized species (O 1s component at BE = 530.6 eV) was found on the surface of the Nb/NbN-HT coated sample, where $SiO_2$ species were found as a contribution from the substrate. A different distribution of oxygen species as NbO and $Nb_2O_5$ was detected on the surface of NbN-RT and NbN-HT samples. As a consequence of the temperature treatment, a relatively higher content of $Nb_2O_5$ species resulted on the surface of the NbN-HT coating.

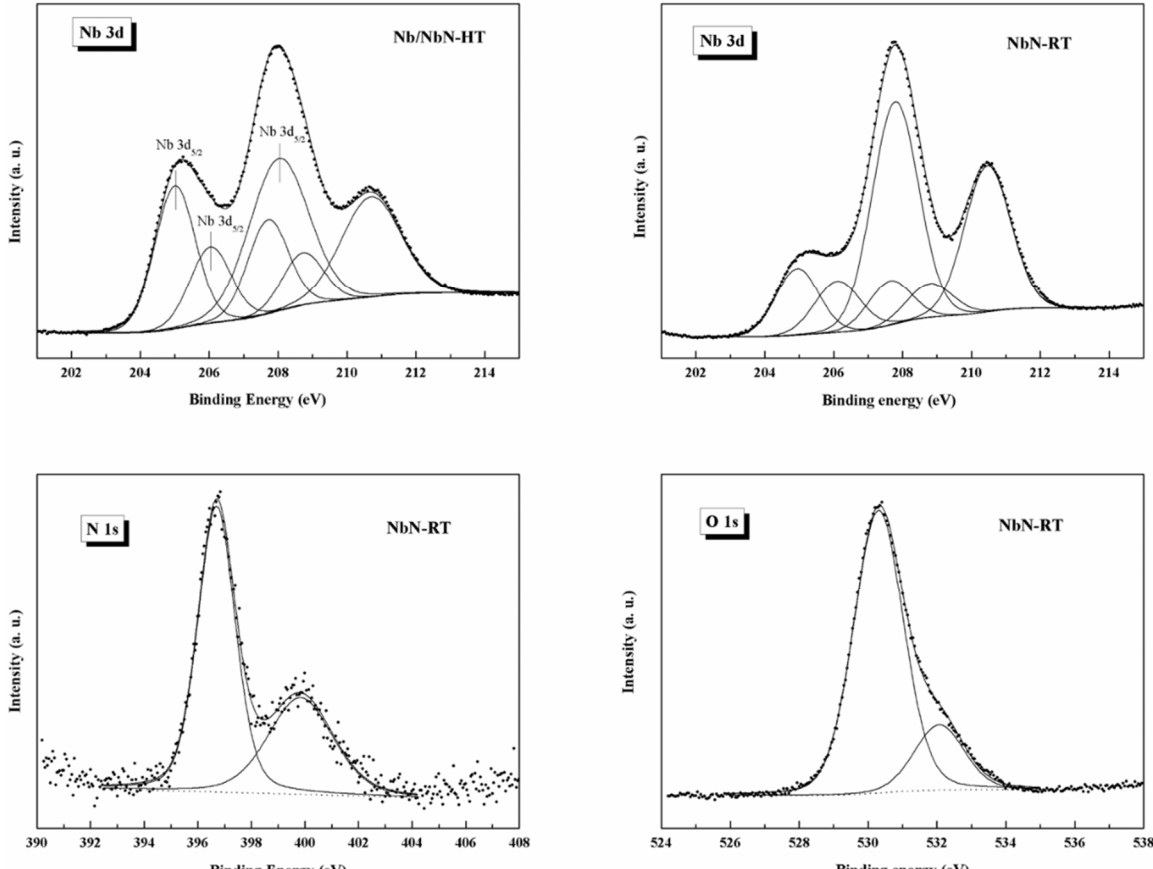

**Figure 6.** XPS curve-fitting of N 3d, N 1s, and O 1s spectra in the investigated samples.

**Table 5.** XPS curve-fitting results of Nb 3d spectra (total peak area = 100%).

| Nb $3d_{5/2}$ BE Assignment | 204.7–205.1 eV $NbN_x$ and/or NbO | 206.2 eV $NbN_xO_y$ | 207.8 eV $Nb_2O_5$ |
|---|---|---|---|
| NbN-RT | 21.8 | 12.1 | 66.1 |
| NbN-HT | 29.6 | 14.8 | 56.3 |
| Nb/NbN-HT | 36.8 | 15.4 | – |

The surface distribution of nitrogen species on the Nb/NbN multilayer coating was extracted from the curve-fitting of N 1s and presented in Table 6 (total peak area = 100%). On the surface of the Nb/NbN-HT coating, the higher concentration of the peak component located at BE = 397.6 eV, which can be assigned to the presence of the Si-N bond, confirmed the influence of the substrate, as found in the case of oxygen distribution (Table 7). A very similar distribution of nitrogen species was detected on the surface of the NbN-RT and NbN-HT coated samples.

**Table 6.** XPS curve-fitting results of N 1s spectra (total peak area = 100%).

| N 1s BE Assignment | 396.6 eV NbN | 397.6 eV NbN/Si-N | 399.8–400.4 eV $NbN_xO_y$ |
|---|---|---|---|
| NbN-RT | 58.6 | 12.2 | 29.3 |
| NbN-HT | 56.8 | 13.6 | 29.6 |
| Nb/NbN-HT | – | 79.1 | 20.9 |

**Table 7.** XPS curve-fitting results of O 1s spectra (total peak area = 100%).

| O 1s BE Assignment | 530.6 eV $O^{2-}$ ions as in NbO | 531.7 eV $Nb_2O_5$ | 532.7399.8–400.4 eV $SiO_2$ |
|---|---|---|---|
| NbN-RT | 72.4 | 27.6 | – |
| NbN-HT | 59.2 | 40.8 | – |
| Nb/NbN-HT | 45.2 | – | 54.8 |

### 3.3. Surface Wettability

The performance of PEMFC is significantly influenced by the wettability behavior of the BPPs. It is well documented that the use of coatings with low wettability or high hydrophobicity for BPPs is beneficial for water removal from the stack [6]. In this work, the surface wettability of the uncoated 304 SS substrate and different coatings was assessed by measuring the contact angle of water drop on surfaces. A high contact angle means low wettability or high hydrophobicity. Figure 7 illustrates the water contact angles recorded for the different specimens. As shown, the hydrophobicity of the substrate was significantly improved by applying the NbN-based coatings (statistically significant ($p < 0.05$). The water contact angles of the examined specimens were arranged as follows: [304 SS] < [Nb-HT] < [NbN-RT] < [Nb/NbN-HT] < [NbN-HT].

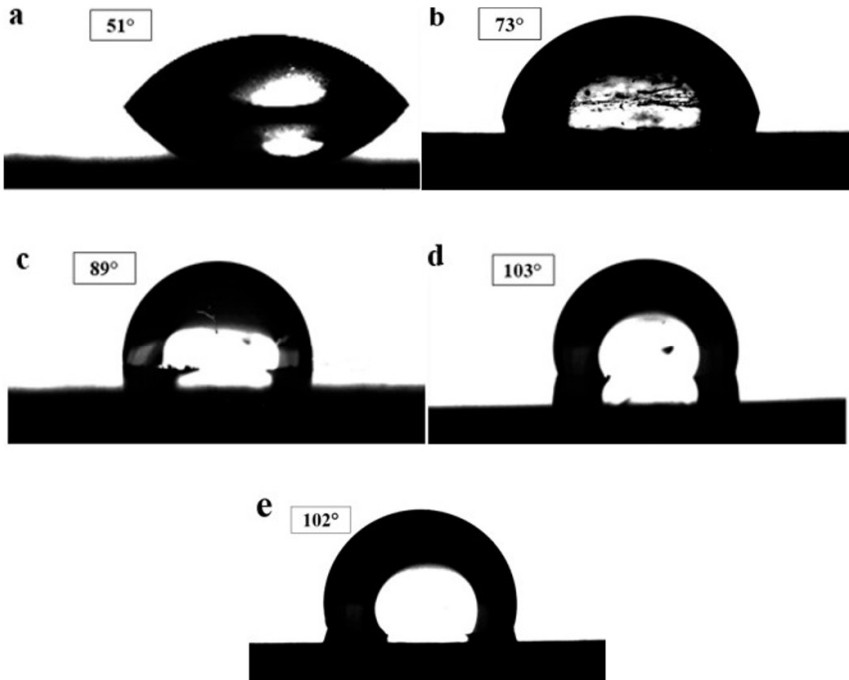

**Figure 7.** Water contact angle of: (**a**) bare 304 SS, (**b**) Nb-HT, (**c**) NbN-RT, (**d**) NbN-HT, (**e**) Nb/NbN-HT.

Among all tested specimens, the deposited NbN and Nb/NbN coatings exhibited the best hydrophobicity. In contrast, the uncoated 304 SS substrate revealed the lowest hydrophobicity. These observations can be related to the fact that the air-formed oxide film on the surface of 304 SS has a

higher affinity for water than the niobium nitride. The high wettability of 304 SS can be due to the presence of oxygen in its surface film (oxygen is more electronegative than niobium and nitrogen).

## 3.4. Electrochemical Measurements

The corrosion resistance of the coatings was assessed using potentiodynamic and potentiostatic polarization tests in simulated PEMFC solution (1 M $H_2SO_4$ + 2 mg/kg HF, 70 °C). As shown in Figure 8, the potentiodynamic polarization curves exhibited a typical active behavior for all the specimens. The electrochemical features as obtained by using Tafel extrapolation are summarized in Table 8. From the results, it can be concluded that the PVD coatings significantly enhanced the corrosion resistance of the 304 SS. Among the tested thin films, the Nb-HT coating exhibited the best corrosion resistance. According to the literature [38,39], the superior corrosion behavior of the Nb-based coatings can be attributed to their higher nobility and greater thermodynamic stability. The high corrosion resistance of the coatings deposited at a high temperature can be related to their homogeneous and compact morphology. Furthermore, the superior hydrophobic property of the NbN-based coatings deposited at high temperature enhanced the removal of the water and improved the anti-corrosion behavior of the specimens.

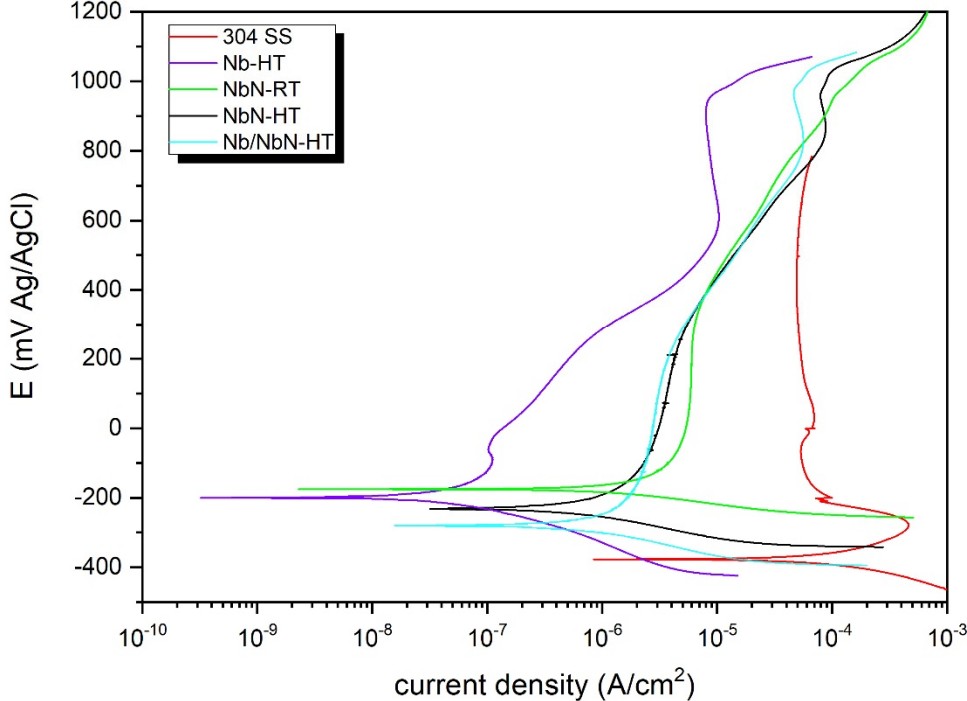

**Figure 8.** Potentiodynamic polarization curves of the uncoated 304 SS and different coatings in 1 M $H_2SO_4$ + 2 mg/kg HF solution at 70 °C.

**Table 8.** Electrochemical features of bare and coated 304 SS tested in simulated PEM fuel cell solution. ($E_{corr}$: corrosion potential and $i_{corr}$: corrosion current density).

| Sample | $E_{corr}$ vs. Ag/AgCl (mV) | $i_{corr}$ ($\mu A/cm^2$) |
|---|---|---|
| 304 SS | −377 ± 45 | 80 ± 12 |
| Nb-HT | −200 ± 23 | 0.045 ± 0.002 |
| NbN-HT | −231 ± 20 | 0.5 ± 0.01 |
| NbN-RT | −175 ± 15 | 1.5 ± 0.05 |
| Nb/NbN-HT | −281 ± 10 | 0.7 ± 0.1 |

The corrosion current density values of different specimens can be arranged as follows: 304 SS uncoated substrate < [PVD-NbN-RT] < [PVD-NbN-HT] < [PVD-Nb/NbN-HT] < [PVD-Nb-HT]

(statistically significant ($p < 0.05$)). The corrosion current densities of the coated 304 SS with Nb-HT, NbN-HT and Nb/NbN-HT were 0.045, 0.5 and 0.7 $\mu A/cm^2$, respectively, which satisfied the US DOE requirement of corrosion resistance (1 $\mu A/cm^2$).

In order to investigate the corrosion and stability of different coatings under anodic and cathodic conditions, the potentiostatic experiments were carried out at typical potentials of real PEMFC working conditions. The experiments were accomplished at $-0.056$ and 0.644 V (vs. Ag/AgCl) to simulate the anodic and cathodic conditions, respectively. Figure 9 illustrates the stability of the coated and uncoated 304 SS, which indicates that the current density is high for both cathodic and anodic conditions and then it decreases continuously versus time and becomes stable after 1800 s. Table 9 presents the stable current densities obtained from potentiostatic tests at different potentials. It indicates that the current densities of the coated specimens at both cathodic and anodic potentials are significantly lower as compared with those of the bare 304 SS (statistically significant ($p < 0.05$)).

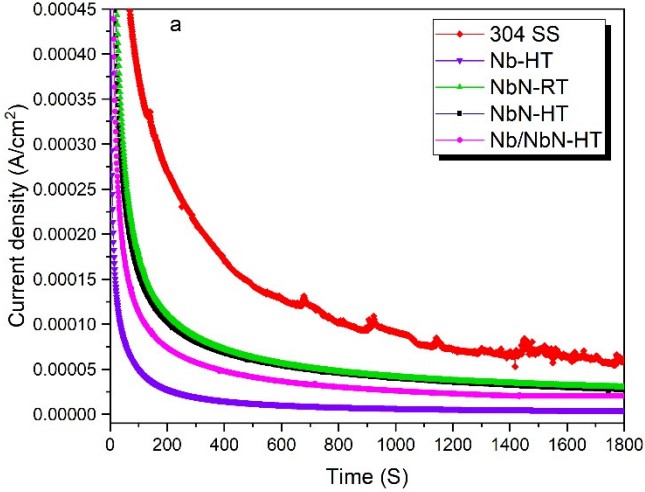

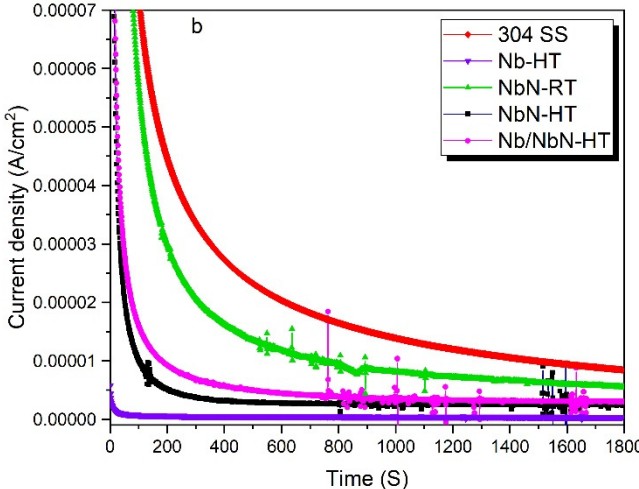

**Figure 9.** Potentiostatic polarization behavior for the uncoated 304 substrate and different coatings in 1 M $H_2SO_4$ + 2 mg/kg HF solution and 70 °C at (**a**) simulated cathode (+0.644 V vs. Ag/AgCl) and (**b**) simulated anode ($-0.056$ V vs. Ag/AgCl).

It needs to be noted that corrosion is rarely observed on the anodic side of PEMFCs. However, the environment on the cathodic side of PEMFC is highly corrosive due to its higher working potential [40]. The potentiostatic polarization tests indicate that the PVD-Nb coated 304 stainless steel

has the lowest current density, which is in accordance with the potentiodynamic results. The decrease in current density of the uncoated 304 SS can be related to the formation of a passive film on its surface. However, its current density was still very high in both simulated anodic and cathodic sides. In addition, the presence of fluctuations in the current density-time of the uncoated substrate in the simulated cathodic side can be attributed to its non- protective and non-stable passive film. All the coated specimens exhibited approximately stable current densities at both cathodic and anodic working potentials of PEMFC. These results indicated that the coatings obtained by RF magnetron sputtering possess good stability and corrosion resistance in a PEMFC environment. It needs to be noted that the appearance of some fluctuations can be related to the presence of pinholes in the coating.

**Table 9.** Stable current density of coated and bare 304 stainless steel after potentiostatic measurements for 30 min.

| Sample | i + 644 mV vs. Ag/AgCl ($\mu$A/cm$^2$) | i − 56 mV vs. Ag/AgCl ($\mu$A/cm$^2$) |
|---|---|---|
| 304 SS | 60.2 ± 12 | 60.5 ± 15 |
| Nb-HT | 4.2 ± 0.8 | 0.4 ± 0.1 |
| NbN-RT | 29.8 ± 4 | 5.5 ± 0.7 |
| NbN-HT | 27.3 ± 5 | 2.5 ± 0.5 |
| Nb/NbN-HT | 20.1 ± 3 | 3.0 ± 0.6 |

### 3.5. Interfacial Contact Resistance Measurement

The conductivity of BPPs is a key factor impacting the PEMFCs' working efficiency. The interfacial contact resistance (ICR) between the gas diffusion layer (GDL) and bipolar plate accounts for the main voltage loss in PEMFC, which can significantly reduce the power density of a fuel cell stack. As a result, low ICR in bipolar plate/GDL interface is a key issue for PEMFC. Table 10 shows the ICR of different specimens recorded at typical compaction force of a single cell (140 N/cm$^2$) [41]. It can be seen that the ICR of the uncoated 304 SS is much larger than that of the coated specimens. The high ICR of the bare 304 SS can be due to the presence of an insoluble surface oxide layer on its surface, which is composed of iron, nickel, and chromium oxides [42]. It also needs to be noted that stainless steels are not intrinsically good electrical conductors. It has been reported that the conductivity of NbN is around $3 \times 10$ $\Omega^{-1}$ cm$^{-1}$ (at 25 °C), which is much higher than that for $Cr_2O_3$ ($<10^{-3}$ $\Omega^{-1}$ cm$^{-1}$) [43,44].

**Table 10.** The interfacial contact resistance of bare and coated 304 SS at compaction force of 140 N/cm$^2$.

| Sample Specimen | i + 644 mV vs. Ag/AgCl ($\mu$AICR in 140 N/cm$^2$ (m$\Omega\cdot$cm$^2$) | i − 56 mV vs. Ag/AgCl ($\mu$A/cm$^2$) |
|---|---|---|
| 304 SS | 60.2 ± 12 | 60.5 ± 15 |
| Nb-HT | 4.2 ± 0.8 | 0.4 ± 0.1 |
| NbN-RT | 29.8 ± 4 | 5.5 ± 0.7 |
| NbN-HT | 27.3 ± 5 | 2.5 ± 0.5 |
| Nb/NbN-HT | 20.1 ± 3 | 3.0 ± 0.6 |

Among all the coated specimens, the Nb coating exhibited the lowest ICR (9 m$\Omega\cdot$cm$^2$), which satisfied the U.S. DOE requirement (≤10 m$\Omega\cdot$cm$^2$) [45]. Moreover, the NbN (NbN-HT) and Nb/NbN coatings prepared at high temperatures showed lower ICR as compared with that of the NbN monolayer coating deposited at low temperatures (NbN-RT). Moreover, the NbN monolayer (PVD-NbN-HT) and PVD-Nb/NbN multilayer coatings prepared at high temperatures showed lower ICR compared with that of the NbN monolayer coating deposited at low temperatures (PVD-NbN-RT). This can be attributed to the higher surface roughness of the NbN-based coatings fabricated at high temperatures (Table 3). It is worthwhile noting that roughness has a strong influence on the ICR values. It has

been demonstrated that for samples with roughness of less than ~1 μm, the rougher surface is more beneficial to ICR reduction [46,47]. In addition, the chemical composition of the coatings is one of the decisive factors influencing the ICR of coatings. The XPS results revealed that the amount of Nb oxidized Nb species of the PVD-NbN coating prepared at low temperatures was the highest among the investigated coatings. Therefore, the higher ICR of the PVD-NbN coating prepared at low temperature can be correlated to the presence of the oxidized species on its surface. Finally, it can be deduced that the ICR values of the NbN-based coatings fabricated at high temperatures are very close to the requirement of the U.S. DOE ($\leq 10$ mΩ·cm$^2$) at 140 N·cm$^2$.

## 4. Conclusions

Nb, NbN, and Nb/NbN thin films were successfully deposited on 304 SS as bipolar plates for PEMFCs by employing an RF magnetron sputtering system. The results indicated that all the deposited coatings exhibited a grain-like morphology. A similar top surface morphology was observed for the NbN coatings deposited at low and high temperatures. Also, it was also found that high substrate temperature increased the roughness of the coatings. According to XRD analyses, higher crystallinity was detected in the NbN coating deposited at high temperatures.

The XPS measurements further revealed that the NbN monolayers prepared at low (room temperature) and high temperatures (350 °C) possessed a very similar surface chemical composition. Furthermore, the lowest and highest concentrations of the Nb oxidized species were found on the surface of the Nb/NbN multilayer and NbN coatings monolayer (deposited at room temperature), respectively. The wettability experiments revealed that all prepared coatings showed higher hydrophobicity in comparison to the uncoated substrate. Among the different coatings, the Nb and NbN (fabricated at high temperature) thin films exhibited the lowest and highest hydrophobicity, respectively. The corrosion assessments using potentiodynamic and potentiostatic evaluations revealed that the Nb-based coated 304 SS exhibited very good corrosion resistance in simulated fuel cell environments and meet the DOE's 2020 technical targets. It was also found that the thin films prepared at high temperatures can provide better corrosion resistance. Potentiostatic polarization tests revealed that the Nb coating exhibited the lowest current density both in anodic and cathodic media environments. The ICR results showed that the surface conductivity of the Nb-based coatings is markedly higher than that for the 304 SS substrate. The lowest amount of ICR was recorded for the Nb coating monolayer (9 mΩ·cm$^2$), which satisfied the DOE requirement of electrical conductivity for the BPPs of PEMFCs. Overall, niobium (PVD-Nb-HT) and niobium nitride PVD coatings prepared at high temperatures (PVD-NbN-HT) could be regarded as promising candidates for BPPs in PEMFCs thanks to their superior properties, such as higher corrosion resistance and wettability, lower ICR, and better long-term stability in cathodic and anodic environments of PEMFCs.

In summary, this work shows that Nb and NbN thin films are highly promising materials for BPPs in PEMFC. However, further studies are required to examine the long-term performance and durability of these thin films for fuel cell applications.

**Author Contributions:** Conceptualization, investigation, validation, formal analysis, methodology, writing—original draft, visualization, M.A.; investigation, formal analysis, writing—original draft, visualization, V.R.; conceptualization, formal analysis, methodology, project administration, visualization, writing—review & editing, S.T.; conceptualization, investigation, writing—review & editing, M.P.C.; conceptualization, investigation, writing—review & editing, G.L.C. All authors have read and agreed to the published version of the manuscript.

**Funding:** This research received no external funding.

**Conflicts of Interest:** The authors declare no conflict of interest.

## Abbreviations

| | |
|---|---|
| PVD | Physical vapor deposition |
| BPP | Bipolar plate |
| PEMFC | Proton-exchange membrane fuel cell |

| RF | Radio-frequency |
| ICR | Interfacial contact resistance |
| SS | Stainless steel |
| TRD | Thermo-reactive diffusion |
| PSDA | Plasma surface diffusion alloying |
| SEM | Scanning electron microscopy |
| XRD | X-ray diffraction |
| XPS | X-ray photoelectron spectroscopy |
| ESD | Energy-dispersive X-ray spectroscopy |
| OCP | Open circuit potential |
| US DOE | United States Department of Energy |

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
