# Peer review of "Thin Niobium and Niobium Nitride PVD Coatings on AISI 304 Stainless Steel as Bipolar Plates for PEMFCs"

_coatings, doi:10.3390/coatings10090889_

Round 1

Reviewer 1 Report

Thin Niobium and Niobium Nitride PVD Coatings on AISI 304 Stainless Steel as Bipolar Plates for 4 PEMFCs by Masoud Atapour Masoud Atapour Chiarello, Vahid Rajaei, Stefano Trasatti, Maria Pia Casalett and Gian Luca Chiarello manuscript is a study of Niobium and Niobium Nitride thin coating produced by Physical Vapour Deposition technique on a AISI 304 stainless steels bipolar plate (BPP) for proton-exchange membrane fuel cell. Four different coatings were investigated, one layer of Nb PVD coating, deposited at 350 °C, one layer of NbN PVD coating, deposited at 25 °C, one layer of NbN PVD coating deposited at 350 °C and two layers of PVD coating consisting of Nb and NbN deposited at 350°C. The coatings were characterized in terms of surface morphology, chemical composition. electrochemical properties and Interfacial contact resistance.

The manuscript has valuable data and it’s very interesting. I have only few minor revisions for the authors:

1. Please check the number of tables. At page 4 there are two table 1 and at page 12 there is for the second time table 5.

2. Please explain better in which way you measured the thickness of coatings.

Author Response

Response to reviewers’ comments

We would like to thank all reviewers for their accurate and valuable comments. We believe that they helped us to significantly improve the manuscript. All changes in the manuscript have been marked, and responses are provided point-by-point below.

Response to Reviewer 1 Comments

Thin Niobium and Niobium Nitride PVD Coatings on AISI 304 Stainless Steel as Bipolar Plates for 4 PEMFCs by Masoud Atapour Masoud Atapour Chiarello, Vahid Rajaei, Stefano Trasatti, Maria Pia Casalett and Gian Luca Chiarello manuscript is a study of Niobium and Niobium Nitride thin coating produced by Physical Vapour Deposition technique on a AISI 304 stainless steels bipolar plate (BPP) for proton-exchange membrane fuel cell. Four different coatings were investigated, one layer of Nb PVD coating, deposited at 350 °C, one layer of NbN PVD coating, deposited at 25 °C, one layer of NbN PVD coating deposited at 350 °C and two layers of PVD coating consisting of Nb and NbN deposited at 350°C. The coatings were characterized in terms of surface morphology, chemical composition, electrochemical properties and Interfacial contact resistance.

The manuscript has valuable data and it’s very interesting. I have only few minor revisions for the authors:

Point 1:  Please check the number of tables. At page 4 there are two table 1 and at page 12 there is for the second time table 5.

Response 1: Many thanks for your positive view on the paper and your accurate reading. We have revised the paper based on your comment.

Point 2:  Please explain better in which way you measured the thickness of coatings.

Response 1: We appreciate for this comment. In order to determine the accurate thickness of the coatings, the coated film was fabricated on a silicon wafer in parallel to the stainless steel substrate. The, the thicknesses were evaluated using a profilometer (Bruker Dektak XT). We have addressed this important point in the revised paper.

Reviewer 2 Report

It is opinion of the reviewer that this paper interesting and valuable from practical point of view before acceptance by “Coatings” needs several corrections/ modifications. My individual comments are listed below.

I suggest addition of the list of abbreviations.

Subscripts should be used for chemical formulas.

22 and other lines. “ppm” is not an unit of SI.

106-111 – The aim of the work should be changed. Now is like summary.

109 – The use power of the ultrasonic bath?

160-161 – The used equipment should be described.

170 – A reference should be added.

A section “Statistical analysis” should be completed.

Tables 2,5, 7 & 8 – The significance of the differences between results should be verified using statistical methods.

References – The abbreviations of journal titles are needed.

Author Response

Response to reviewers’ comments

We would like to thank all reviewers for their accurate and valuable comments. We believe that they helped us to significantly improve the manuscript. All changes in the manuscript have been marked, and responses are provided point-by-point below.

Response to Reviewer 2 Comments

Comments and Suggestions for Authors

It is opinion of the reviewer that this paper interesting and valuable from practical point of view before acceptance by “Coatings” needs several corrections/ modifications. My individual comments are listed below.

We wish to express our appreciation to the Reviewer for positive views and insightful comments, which have helped us significantly to improve our manuscript.

Point 1: I suggest addition of the list of abbreviations.

Response 1: Many thanks for this comment. We have added the list of abbreviations in the revised paper.

List of abbreviations:

PVD: Physical vapor deposition

BPP: Bipolar plate

PEMFC: Proton-exchange membrane fuel cell

RF: Radio-frequency 

ICR: Interfacial contact resistance

SS: Stainless steel

TRD: Thermo-reactive diffusion

PSDA: Plasma surface diffusion alloying

SEM: Scanning electron microscopy

XRD: X-ray diffraction

XPS: X-ray photoelectron spectroscopy

ESD: Energy-dispersive X-ray spectroscopy

OCP: Open circuit potential

US DOE: United States Department of Energy

Point 2: Subscripts should be used for chemical formulas.

Response 2: We have modified the paper based on this comment.

Point 3: 22 and other lines. “ppm” is not an unit of SI.

Response 3: We have corrected the units based on this comment.

Point 4: 106-111 – The aim of the work should be changed. Now is like summary.

Response 4: Thank you for this comment. We have addressed this comment in the revised paper. 

Point 5: 109 – The use power of the ultrasonic bath?

Response 5: It has been corrected in the revised version.

Point 6: 160-161 – The used equipment should be described.

Response 6: We have described the equipment according to this comment.

Point 7: 170 – A reference should be added.

Response 7: We have added a reference based on this comment.

Point 8: A section “Statistical analysis” should be completed.

Response 8: We have completed the statistical analysis.

Point 9: Tables 2,5, 7 & 8 – The significance of the differences between results should be verified using statistical methods.

Response 9: We have conducted at least three replicates for all experiments. We have addressed this comment.

Point 10: References – The abbreviations of journal titles are needed.

Response 10: We have modified the references according this comment.

Once again, we thank the Editor and Reviewers for a job well done, and hope that the paper will be accepted for publication in Coating.

Reviewer 3 Report

The authors have sputtered Nb, NbN, and Nb/NbN films on top of AlSI 304 substrate. The targeted application is as bipolar plate for fuel cell. Therefore, in addition to the standard characterization tests of SEM, XRD, and XPS, surface rougness, and thickness, they also performed electrochemical and contact resistance measurement to determine the suitability as anode and cathode plates. 

The manuscript is clearly written, and the investigation is complete. However, I could not recommend it for publication in present form because it does not have sufficient novelty. The use of Nb and NbN sputtered on AlSI 304 as bipolar plates for fuel cell has been published widely: https://scholar.google.com/scholar?hl=id&as_sdt=0%2C5&q=nb+nbn+304+fuel+cell&btnG=

Further, the authors claim in line 108 that there is no prior work on the magnetron sputtering of NbN on top of AlSI 304. This is not true, as shown by this Korean paper: https://www.researchgate.net/profile/Junho_Kim5/publication/264170269_Characteristics_of_NbN_Films_Deposited_on_AISI_304_Using_Inductively_Coupled_Plasma_Assisted_DC_Magnetron_Sputtering_Method/links/5428a64c0cf238c6ea7cd882.pdf

This is a good work, but with the current narrative, has low impact. I suggest the authors to redo the literature review, and then rewrite the manuscript to highlight their contribution. 

Author Response

Manuscript ID: coatings-899993
Type of manuscript: Article
Title: Thin niobium and niobium nitride PVD coatings on AISI 304 stainless
steel as bipolar plates for PEMFCs

Response to reviewers’ comments

We would like to thank all reviewers for their accurate and valuable comments. We believe that they helped us to significantly improve the manuscript. All changes in the manuscript have been marked, and responses are provided point-by-point below.

Response to Reviewer 1 Comments

The authors have sputtered Nb, NbN, and Nb/NbN films on top of AlSI 304 substrate. The targeted application is as bipolar plate for fuel cell. Therefore, in addition to the standard characterization tests of SEM, XRD, and XPS, surface rougness, and thickness, they also performed electrochemical and contact resistance measurement to determine the suitability as anode and cathode plates.

The manuscript is clearly written, and the investigation is complete. However, I could not recommend it for publication in present form because it does not have sufficient novelty. The use of Nb and NbN sputtered on AlSI 304 as bipolar plates for fuel cell has been published widely: https://scholar.google.com/scholar?hl=id&as_sdt=0%2C5&q=nb+nbn+304+fuel+cell&btnG=

Further, the authors claim in line 108 that there is no prior work on the magnetron sputtering of NbN on top of AlSI 304. This is not true, as shown by this Korean paper: https://www.researchgate.net/profile/Junho_Kim5/publication/264170269_Characteristics_of_NbN_Films_Deposited_on_AISI_304_Using_Inductively_Coupled_Plasma_Assisted_DC_Magnetron_Sputtering_Method/links/5428a64c0cf238c6ea7cd882.pdf

This is a good work, but with the current narrative, has low impact. I suggest the authors to redo the literature review, and then rewrite the manuscript to highlight their contribution.

We wish to express our appreciation to the Reviewer for positive views and insightful comments, which have helped us significantly to improve our manuscript.

Response 1: Many thanks for this comment. As you mentioned, there are many works on NbN coatings fabricated by different techniques for various applications. We have addressed many of them in introduction. However, limited works have been reported on developing the Nb and NbN coatings produced by RF magnetron sputtering PVD method for use in bipolar plates of Fuel Cell. In addition, the outstanding properties of the Nb and NbN coatings deserve to be studied deeper for use in bipolar plates.

The main novelty of this work is to develop the Nb and NbN monolayers as well as Nb/NbN multilayer coatings for BPPs applications on 304 SS by employing a radio frequency (RF) magnetron sputtering system.

We have modified the paper based on this comment.

Once again, we thank the Editor and Reviewers for a job well done, and hope that the paper will be accepted for publication in Coating.

Yours sincerely

Masoud Atapour

Round 2

Reviewer 3 Report

The authors have rewritten the last part of section 1 to emphasizing the novelty in term of testing Nb/NbN composite as BPP and additional tests. These are very small and incremental contributions, but is publishable in an open access journal.

There are typos such as 5o W etc, but this can be corrected by proff team.

Author Response

The authors have rewritten the last part of section 1 to emphasizing the novelty in term of testing Nb/NbN composite as BPP and additional tests. These are very small and incremental contributions, but is publishable in an open access journal.

Response: Many thanks for your valuable comments. We have again modified this section based on you comment.

There are typos such as 5o W etc, but this can be corrected by proff team.

Response: We have corrected the typos in the revised paper.